# Efficacy of S-1 or Capecitabine Plus Oxaliplatin Adjuvant Chemotherapy for Stage II or III Gastric Cancer after Curative Gastrectomy: A Systematic Review and Meta-Analysis

**DOI:** 10.3390/cancers14163940

**Published:** 2022-08-16

**Authors:** Sang-Ho Jeong, Rock Bum Kim, Sung Eun Oh, Ji Yeong An, Kyung Won Seo, Jae-Seok Min

**Affiliations:** 1Department of Surgery, Gyeongsang National University Changwon Hospital, Gyeongsang National University College of Medicine, Changwon 51472, Korea; 2Regional Cardiocerebrovascular Disease Center, Gyeongsang National University Hospital, Jinju 52727, Korea; 3Department of Surgery, Samsung Medical Center, Sungkyunkwan University School of Medicine, Seoul 06351, Korea; 4Department of Surgery, Kosin University Gospel Hospital, Busan 49267, Korea; 5Department of Surgery, Dongnam Institute of Radiological and Medical Sciences, Cancer Center, Busan 46033, Korea

**Keywords:** gastric neoplasm, adjuvant chemotherapy, S-1, CAPOX, survival

## Abstract

**Simple Summary:**

Adjuvant chemotherapy regimens tegafur/gimeracil/oteracil (S-1) and capecitabine plus oxaliplatin (CAPOX) have predominated, owing to evidence of their remarkable oncologic outcomes, however, there has been a lack of studies on the difference in efficacy between the two regimens. We conducted pairwise meta-analyses comparing the efficacy of S-1 and CAPOX regimens for overall survival (OS) and disease-free survival (DFS) in stage II or III gastric cancer patients. In all stages (stages II and III), the five-year OS was not different between the two regimens (hazard ratio [HR] 0.96, 95% confidence interval [CI] 0.78–1.17; *p* = 0.56). Additionally, the five-year DFS was not different at any stage (HR 1.00, 95% CI 0.85–1.18; *p* = 0.21). The present meta-analysis showed that five-year OS and DFS for stage II or III gastric cancer patients were comparable between the S-1 and CAPOX adjuvant chemotherapy regimens.

**Abstract:**

**Background:** Adjuvant chemotherapy (AC) regimens tegafur/gimeracil/oteracil (S-1) and capecitabine plus oxaliplatin (CAPOX) have predominated, however, there has been a lack of studies on their differences in efficacy. **Methods:** We conducted pairwise meta-analyses comparing the efficacy of S-1 and CAPOX regimens for overall survival (OS) and disease-free survival (DFS) in stage II or III GC patients. **Results:** Three studies were enrolled and analyzed using a forest plot for meta-analysis. Two of them were propensity score matching studies, and the remaining one was a retrospective observational study. In all stages, the five-year OS was not different between the two regimens (HR 0.96, 95% CI 0.78–1.17; *p* = 0.56). Additionally, the 5-year DFS was not different at any stage (HR 1.00, 95% CI 0.85–1.18; *p* = 0.21). After omitting the retrospective observational study, the five-year OS (HR 1.40, 95% CI 0.53–3.73) and DFS (HR 1.41, 95% CI 0.57–3.44) of S-1 tended to be better in stage II, and the five-year OS (HR 0.81, 95% CI 0.56–1.16) and DFS (HR 0.85, 95% CI 0.63–1.13) of CAPOX tended to be better in stage III, without statistical significance. **Conclusions**: In the present meta-analysis, the five-year OS and DFS for stage II or III GC patients were comparable between S-1 and CAPOX regimens as AC.

## 1. Introduction

Patients with locally advanced gastric cancer (AGC) are recommended to undergo additional treatment after curative gastrectomy for stage II or III gastric cancer [1,2,3,4,5]. Worldwide, there are various adjuvant strategies based on different surgical procedures for AGC in different countries [6]. In the West, postoperative chemoradiation therapy or perioperative chemotherapy has been commonly performed, while D2 lymph node dissection (LND) is less frequently performed for AGC [7,8,9,10]. On the other hand, in the East, the adjuvant treatment is postoperative chemotherapy, because curative surgical resection with D2 LND is the standard treatment for AGC in Asia [1,2,5]. However, in the West, curative gastrectomy with D2 LND is recommended based on the updated findings of a Dutch trial, in which locoregional recurrence and gastric cancer-related death rates were found to be lower 15 years after D2 LND [11]. Thereafter, the necessity of adjuvant chemotherapy (AC) has been recognized as important.

As AC regimens, tegafur/gimeracil/oteracil (S-1) and capecitabine plus oxaliplatin (CAPOX) have predominated, owing to their remarkable oncologic outcomes in the randomized controlled trials ACTS-GC and CLASSIC [12,13]. The trial of S-1, ACTS-GC, confirmed the survival efficacy of adjuvant oral S-1 monotherapy [12]. The CLASSIC trial reported that adjuvant CAPOX improved disease-free survival compared with surgery alone [13]. However, it has not been possible to compare the accurate survival differences of the two AC regimens based on the results of two different studies. In actual clinical situations, oncologists are concerned about which AC regimen to choose after curative gastrectomy.

In recent years, several studies have directly compared the efficacies of adjuvant S-1 and CAPOX regimens for AGC. Among the studies comparing three-year postoperative survival, a multicenter propensity score matching (PSM) study showed that adjuvant CAPOX chemotherapy was more effective than S-1 for stage IIIB or IIIC gastric cancer [14]. On the other hand, among the studies analyzing five-year postoperative survival, two studies that conducted PSM showed comparable oncologic outcomes between the two regimens [15,16]. A single-center, observational study presented the potential for superior efficacy of the CAPOX regimen against stage II gastric cancer [17].

As such, the results of recent studies are not consistent, and to date, there has been no published prospective study or meta-analysis comparing the efficacy of the S-1 and CAPOX AC regimens. The aim of this meta-analysis was to compare the overall survival (OS) and disease-free survival (DFS) of the S-1 and CAPOX regimens as AC, focusing on oncologic outcomes at five years after curative gastrectomy for AGC.

## 2. Methods

### 2.1. Search Scheme & Selection of Studies

The flow diagram of the meta-analysis search scheme is shown in Figure 1. Three keywords, “gastric neoplasm,” “gastrectomy,” and “chemotherapy,” were used to search the PubMed, Embase, Web of Science, Google Scholar, and Cochrane Library databases (2000–2022). The detailed search keywords included in PubMed and Cochrane Library were as follows: ((gastric OR stomach) AND neoplasms [MeSH])) AND ((operat* OR surg* OR gastrectomy [MeSH])) AND ((drug therapy [MeSH] OR chemotherapy*)). Those in Embase were as follows: ‘stomach disease’/exp AND ‘neoplasm’/exp AND (‘surgery’/exp OR ‘gastrectomy’/exp) AND (‘chemotherapy’/exp OR ‘drug therapy’/exp) AND (‘cancer recurrence’/exp OR ‘death’/exp).

The electronic database searches identified 18,621 papers by hand search. In total, 18,621 studies were identified in November 2021. Among these, we excluded 6741 studies since they were reviews, books, or meta-analyses, and we included 11,880 clinical trials and randomized controlled studies. Based on the titles and abstracts of the 11,875 studies, five clinical studies comparing S-1 versus CAPOX as AC in patients undergoing gastrectomy were selected [14,15,16,17,18]. Two of these studies were excluded because they collected data from a relatively short-term three-year follow-up after gastrectomy [14,18]. The remaining three studies were analyzed using a forest plot, and the results are reported herein [15,16,17].

### 2.2. Study Quality Assessment

We assessed the quality of the three selected studies by using the Newcastle Ottawa Quality Assessment Scale (NOS). The NOS was developed to evaluate the quality of nonrandomized studies, including case–control and cohort studies, in the interpretation of meta-analytic results, based on three categories: group selection (four items), comparability between the groups (one item), and the ascertainment of either the exposure or outcome of interest (three items). Each item could be awarded a maximum of one point in the group selection and exposure/outcome categories and two points in the comparability category. The possible total score ranged from zero to nine, and studies with more than six points were considered good quality and consistent in general.

The assessment of the three selected studies was performed by two independent reviewers (S-H Jeong and RB Kim). After review, any discrepancies between the two reviewers’ opinions were resolved by discussion.

### 2.3. Statistical Analysis of Data

We performed pairwise meta-analyses comparing the S-1 and CAPOX regimens for OS and DFS in stage II gastric cancer, stage III gastric cancer, and all stages (II & III) of gastric cancer (Figure 2). The between-group effect size was computed by calculating the pooled hazard ratio (HR) and 95% confidence interval (CI). The staged OS rates were not presented in two of the previous papers; therefore, OS in previous studies were analyzed by the authors in previous papers, and then the staged OS data were included in this meta-analysis [16,17].

Effect sizes were pooled using common- or random-effects models with a generic invariance method to incorporate the heterogeneity of differences across the studies. The between-study quantification of heterogeneity was measured using I-square statistics, and heterogeneity was tested using Cochran’s Q. As a result, we adopted a random-effects model when the I-square values were more than 50% and the *p* value of Cochran’s Q was less than 0.1; otherwise, a fixed-effects model was adopted. Sensitivity analyses were performed by serially excluding each study (leave-one-out method) to assess the implications of each study on the pooled effect size (Figure 3). Publication bias was assessed using funnel plots, and tested for asymmetry using Egger’s test (Appendix A). All statistical analyses were performed using the metafor (meta-regression) and mice (multiple imputation by chained equations) packages in R software, version 4.0.2 (R Foundation for Statistical Computing, Vienna, Austria). Statistical tests were two-sided for the HRs and one-sided for Egger’s test, with a significance threshold of *p* < 0.05.

## 3. Results

### 3.1. Literature Search and Quality of the Selected Studies

A summary of the three studies is shown in Table 1. In 2019, Shin et al. presented a single-center PSM study that included 110 patients assigned to each regimen [15]. In 2020, Lee et al. conducted a PSM multicenter cohort study with 1:3 matching between the S-1 and CAPOX groups, using the nearest-neighbor matching method [16]. In 2021, Oh et al. performed a retrospective large-scale single-center observational study [17]. The median follow-up periods of the three studies were 52.3 months, 59.0 months, and 55.0 months, respectively.

Appendix A shows the results of the quality assessment of the three selected studies. The respective quality rating scores of the three studies [15,16,17] were 8, 8, and 7. Therefore, the quality of the three studies was considered good.

### 3.2. 5-Year OS of S-1 and CAPOX

The data from the meta-analysis for the five-year OS and DFS of the S-1 and CAPOX regimens for stage II and III gastric cancers are presented in Figure 2. In all stages, the five-year OS was not different between the two regimens (HR 0.96, 95% CI 0.78–1.17; *p* = 0.56) (Figure 2A). When analyzed by dividing patients into stages II and III, the five-year OS of the CAPOX regimen tended to be better in stage II patients, without statistical significance (HR 0.65, 95% CI 0.38–1.11; *p* = 0.14); however, the five-year OS of the two regimens was similar in stage III patients (HR 0.95, 95% CI 0.76–1.19; *p* = 0.50) (Figure 2B,C).

The single-center PSM study by Shin et al. [15] showed a tendency towards a better five-year OS for CAPOX in stages II and III and all stages (Figure 2A–C). On the other hand, the multicenter PSM study by Lee et al. [16] showed a better five-year OS for S-1 (HR 1.66, 95% CI 0.57–4.85) in stage II and for CAPOX (HR 0.86, 95% CI 0.54–1.36) in stage III. The large-scale single-center observational study by Oh et al. [17] reported a significantly better five-year OS for CAPOX (HR 0.47, 95% CI 0.25–0.89) in stage II; however, it resulted in similar OS scores for the two regimens (HR 1.04, 95% CI 0.79~1.38) in stage III.

In the sensitivity analysis of the three enrolled studies, excluding the observational study by Oh et al. [17], CAPOX tended to show a better five-year OS than S-1 in all stages (HR 0.88, 95% CI 0.63–1.23) (Figure 3A). Additionally, in the subgroup analysis by stage, S-1 tended to show a better five-year OS in stage II (HR 1.40, 95% CI 0.53–3.73), and CAPOX tended to show a better five-year OS in stage III (HR 0.81, 95% CI 0.56–1.16), after excluding the observational study by Oh et al. [17] (Figure 3B,C).

### 3.3. 5-Year DFS of S-1 and CAPOX

According to meta-analysis, the five-year DFS for all stages did not differ between the two regimens (HR 1.00, 95% CI 0.85–1.18; *p* = 0.21) (Figure 2D). Shin et al. [15] showed a better five-year DFS for CAPOX in all stages (HR 0.65, 95% CI 0.39–1.09). However, there was no difference in the five-year DFS between the two regimens in the studies by Lee et al. [16] (HR 1.01, 95% CI 0.73–1.40), nor Oh et al. [17]. (HR 1.07, 95% CI 0.87–1.33).

When the subgroup analysis was performed for stage II or III, the five-year DFS was similar to the five-year OS (Figure 2E,F). The single-center PSM study by Shin et al. [15] tended to show a better five-year DFS for CAPOX in stages II (HR 0.35, 95% CI 0.04–3.20) and III (HR 0.67, 95% CI 0.40–1.13). In the multicenter PSM study by Lee et al. [16], S-1 chemotherapy showed a trend toward a better five-year DFS in stage II (HR 1.85, 95% CI 0.69–4.92); however, there was no difference in the five-year DFS between the two regimens in stage III (HR 0.94, 95% CI 0.66–1.34). The large-scale single-center observational study by Oh et al. [17] reported a better five-year DFS for CAPOX in stage II (HR 0.73, 95% CI 0.43–1.24); however, it tended to show a better five-year DFS for S-1 in stage III (HR =1.19, 95% CI 0.92–1.53).

In the sensitivity analysis of the enrolled studies, when we omitted the multicenter PSM study by Lee et al. [16], the five-year DFS was similar between the two regimens in all stages (HR =1.00, 95% CI 0.82–1.22) (Figure 3D). When we omitted the single-center PSM study by Shin et al. [15], a slight trend towards a better five-year DFS for S-1 was observed (HR =1.05, 95% CI 0.88–1.26). If we excluded the observational study by Oh et al. [17], the results showed a better five-year DFS for CAPOX in all stages (HR =0.89, 95% CI 0.68–1.17). After omitting the study by Oh et al. [17], the HR value for five-year DFS in stage II showed a tendency to be better for S-1 (HR =1.41, 95% CI 0.57–3.44), which was the opposite result compared to when either of the other two studies were omitted (Figure 3E). In stage III, after omitting the study by Oh et al. [17], the HR value for five-year DFS tended to be better for the CAPOX regimen (HR =0.85, 95% CI 0.63–1.13), which was the opposite finding compared to when either other study was omitted (Figure 3F).

## 4. Discussion

The present study performed a meta-analysis of recent studies that reported the five-year OS and DFS for the S-1 and CAPOX regimens as AC after curative gastrectomy for stage II or III gastric cancer. In general, the five-year OS and DFS were comparable between the S-1 and CAPOX regimens.

AC is performed as an additional treatment after curative gastrectomy for AGC. In contrast to palliative chemotherapy, AC aims to prevent the recurrence of gastric cancer after curative resection. AC is provided in the interest of enhancing the eradication of microscopic malignant lesions after curative gross resection, and is also proposed as an option for patients with a substantial risk of AGC recurrence [19]. The efficacy of each AC regimen was previously proven in two prominent randomized controlled trials (RCTs). The ACTS-GC trial of S-1 was conducted in Japan, and the CLASSIC trial was performed in Korea, China, and Taiwan [12,13]. Survival outcomes were reported in the ACTS-GC and CLASSIC trials. The efficacy of each adjuvant chemotherapy regimen compared to surgery alone was analyzed in two previous RCTs. We performed this meta-analysis study to compare the differences in oncologic efficacy between S-1 and CAPOX regimens. We did not include the previous two RCTs in this meta-analysis, because the efficacy of the two regimens (S-1 vs CAPOX) was not subjected to a direct head-to-head comparison in the previous two RCTs. To conduct the meta-analysis comparing the efficacy of the two regimens, the data on the differences in oncologic outcomes between S-1 and CAPOX regimens were essential. Furthermore, the stage classification was different in two previous RCTs; the ACTS-GC trial applied the guidelines of the Japanese Gastric Cancer Association, and the CLASSIC trial applied the guidelines of the American Joint Committee on Cancer and Union Internationale Contre le Cancer. Therefore, the composition of enrolled patients was different in the two RCTs, rendering it impossible to perform a direct comparison of data from the ACTS-GC and CLASSIC trials. For these reasons, we did not include two of the previous RCTs in this comparative meta-analysis. One of the systematic reviews compared the performance of a combination regimen over single-agent chemotherapy as an adjuvant treatment for gastric cancer [20]. It was reported that adjuvant combination chemotherapy decreases the risk of death compared to single-agent therapy in patients with nonmetastatic gastric cancer. However, evidence of the different efficacies of the S-1 and CAPOX regimens remains deficient.

Several studies used PSM-applied cohorts or retrospective observational designs directly to compare the efficacy of the S-1 and CAPOX regimens. Kim et al. conducted a PSM-applied multicenter cohort study with a 33.3-month median follow-up period [14]. They compared three-year DFS between two AC regimens, and showed the superior oncologic efficacy of the adjuvant CAPOX regimen in stage IIIB or IIIC AGC. Additionally, Cho et al. analyzed the three-year survival outcomes in a single-center observational study with a 21.0-month median follow-up duration, and found that the adjuvant S-1 and CAPOX regimen outcomes did not significantly differ for stage III gastric cancer [18]. However, Cho et al. concluded that CAPOX tended to be superior to S-1 in stage IIIC gastric cancer, without statistical significance. As such, it is known that the CAPOX regimen seems to be more effective than S-1 in stage III at a relatively short time point of three years after surgery. However, studies comparing the two regimens with a longer follow-up period of five years after gastrectomy reported somewhat different results. Shin et al. compared using PSM the five-year OS and DFS of the two regimens, with a 52.3-month median follow-up period in a single-center study, and found no significant difference in survival between S-1 and CAPOX in stage II and III gastric cancer [15]. Additionally, Lee et al. performed a propensity score-matched multicenter cohort study with a median follow-up period of 59.0 months, finding no significant difference in the five-year DFS outcomes for the two regimens [16]. To summarize the five-year follow-up survival results of PSM studies, the S-1 and CAPOX regimens showed comparable oncologic efficacy.

Postoperative AC treatment is recommended according to guidelines in many countries. Particularly in Europe, Japan, China, and Korea, the guidelines recommend S-1 and CAPOX as AC regimens [21]. Nevertheless, all three studies enrolled in this meta-analysis were conducted in Korea. The reason for this is that the S-1 and CAPOX regimens are provided in Korea under national insurance as an AC drug after curative gastrectomy for stage II or III gastric cancer. Therefore, both regimens have been prescribed for all stage II or III gastric cancers in Korea since national insurance coverage began, based on the past two randomized controlled trials [12,13]. On the other side, the insurance coverage situation for AC treatment is different in other countries. Therefore, all studies to date that have compared the S-1 and CAPOX regimens have been conducted in Korea. Among these, relatively small numbers of patients were enrolled in the studies by Shin et al. (*n* = 110) and Lee et al. (*n* = 155) for the CAPOX regimen, compared to the study by Oh et al. (*n* = 634) [15,16,17]. Nevertheless, Shin et al. and Lee et al. conducted PSM analysis to reduce bias; therefore, they presented more reliable data compared to Oh et al., who performed a single-center retrospective observational study.

The five-year OS rates of the patients receiving the two treatment regimens in all stages of GC were comparable in this meta-analysis. However, although statistically insignificant, the CAPOX regimen was better in stage II patients (HR = 0.65, 95% CI 0.38–1.11) and the results were clinically significant (approximately 1.5 times more deaths in patients who received S-1). Among the three studies included, Oh et al. suggested that the five-year OS of the CAPOX regimen was significantly better than that of S-1 in stage II [17]. However, the other two studies did not show a difference in the five-year OS between the two regimens. Oh et al. designed a retrospective observational study; therefore, the patients in the CAPOX group had a more advanced stage of gastric cancer than those in the S-1 group, suggesting selection bias. On the other hand, Shin and Lee et al. conducted PSM analyses that minimized the influence of potential confounders on selection bias, using a retrospective design [15,16]. When omitting the single-center observational study by Oh et al., the five-year OS (HR 1.40, 95% CI 0.53–3.73) and DFS (HR 1.41, 95% CI 0.57–3.44) of S-1 tended to be better in stage II, and the five-year OS (HR 0.81, 95% CI 0.56–1.16) and DFS (HR 0.85, 95% CI 0.63–1.13) of CAPOX tended to be better in stage III (Figure 3).

This meta-analysis has its limitations. First, few previous studies have compared the efficacy of the S-1 and CAPOX regimens. Second, the three studies enrolled in this meta-analysis included no prospective study. Two studies used PSM in their design, and the other was a retrospective observational study. Third, the present meta-analysis was unable to show the comparative data for toxicity or adverse events for the two AC regimens, because the three papers included did not report data of adverse events. Fourth, the present meta-analysis did not reflect the results of other recent studies, which indicated that response to anti-cancer treatment may be linked to specific population--related genetic variants, referring to the use of molecular profiling in the design of personalized treatment [22]. Fifth, the present meta-analysis did not include other regimens for adjuvant chemotherapy, such as adjuvant S-1 plus docetaxel in stage III gastric cancer or S-1 plus oxalipatin with radiotherapy in stage II or III gastric cancer [23,24]. Nevertheless, to the best of our knowledge, in the current absence of a prospective comparison study, this is the first meta-analysis to compare the survival outcomes of the S-1 and CAPOX adjuvant regimens for AGC.

In conclusion, in the present meta-analysis, the five-year OS and DFS were comparable between the S-1 and CAPOX regimens as AC after curative gastrectomy for stage II or III gastric cancer. 

## Figures and Tables

**Figure 1 cancers-14-03940-f001:**
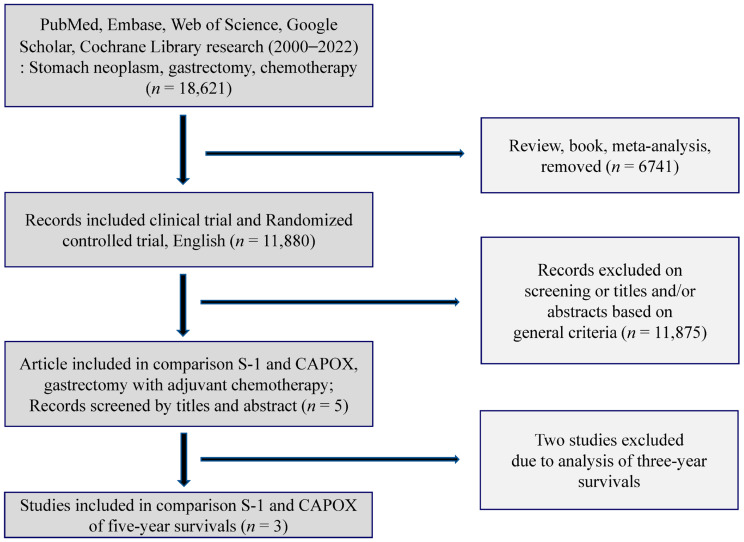
Flow chart of study selection. S-1,tegafur/gimeracil/oteracil; CAPOX, capecitabine plus oxaliplatin.

**Figure 2 cancers-14-03940-f002:**
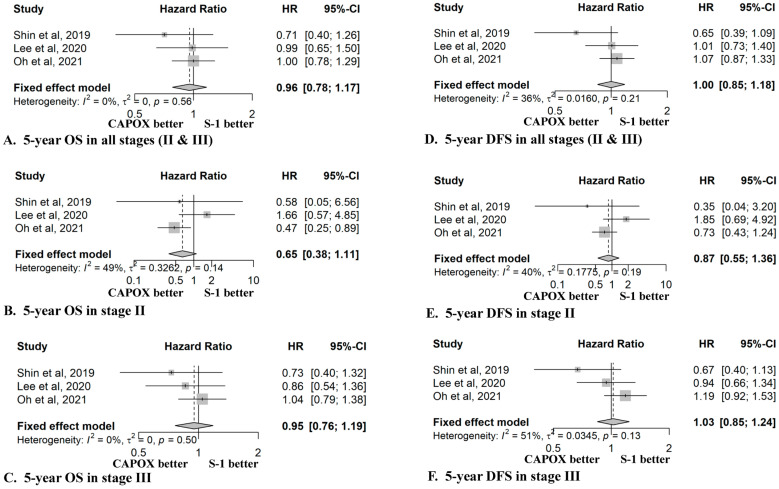
Comparison of the five-year overall survival and disease-free survival for S-1 and CAPOX adjuvant chemotherapy in stage II or III gastric cancer. (**A**) Five-year OS in all stages (II & III), (**B**) five-year OS in stage II, (**C**) five-year OS in stage III, (**D**) five-year DFS in all stages (II & III), (**E**) five-year DFS in stage II, (**F**) five-year DFS in stage III. (**B**) S-1,tegafur/gimeracil/oteracil; CAPOX, capecitabine plus oxaliplatin; HR, hazard ratio; CI, confidence interval; OS, overall survival; DFS, disease-free survival. Shin et al. [15]; Lee et al. [16]; Oh et al. [17].

**Figure 3 cancers-14-03940-f003:**
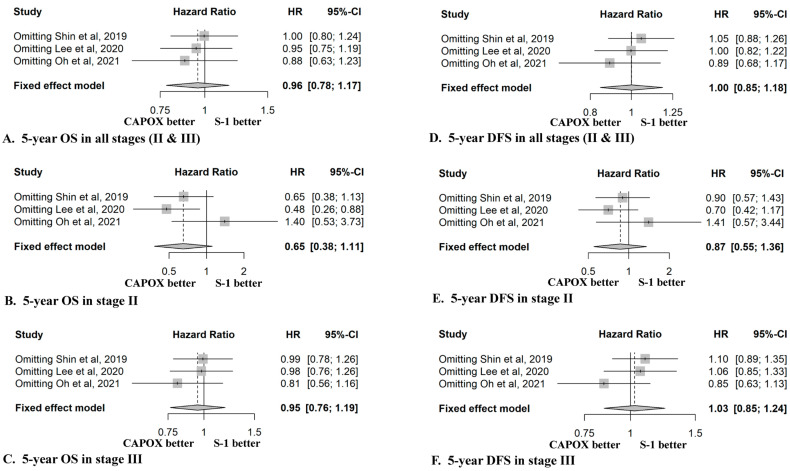
Sensitivity analysis of studies comparing S-1 and CAPOX as adjuvant chemotherapy in stage II or III gastric cancer. (**A**) Five-year OS in all stages (II & III), (**B**) five-year OS in stage II, (**C**) five-year OS in stage III, (**D**) five-year DFS in all stages (II & III), (**E**) five-year DFS in stage II, (**F**) five-year DFS in stage III. (**B**) S-1,tegafur/gimeracil/oteracil; CAPOX, capecitabine plus oxaliplatin; HR, hazard ratio; CI, confidence interval; OS, overall survival; DFS, diseasefree survival. Shin et al. [15]; Lee et al. [16]; Oh et al. [17].

**Table 1 cancers-14-03940-t001:** Summary of studies and HRs of S-1 compared with CAPOX as reference for overall survival and disease-free survival with gastric cancer.

1st Author	Year of Publication	No of Participating Institutes	Study Design	Median Follow-Up Periods (Months)	TNM Stage	No of S-1 Cases	No of CAPOX Cases	5 yr OS HR (S-1 vs. CAPOX, 95% CI)	*p* Value	5 yr DFS, HR (S-1 vs. CAPOX, 95% CI)	*p* Value
Shin et al. [15]	2019	1	PSM	52.3	II & III	110	110	0.71 (0.40–1.26)	0.240	0.65 (0.39–1.09)	0.101
					II	24	23	0.58 (0.05–6.40)	0.655	0.35 (0.04–3.34)	0.360
					III	86	87	0.73 (0.40–1.31)	0.285	0.67 (0.40–1.13)	0.133
Lee et al. [16]	2020	27	PSM	59.0	II & III	429	155	0.986 (0.647–1.504)	0.949	1.008 (0.728–1.395)	0.963
					II	143	50	1.662 (0.569–4.850)	0.353	1.846 (0.693–4.919)	0.220
					III	286	105	0.859 (0.542–1.361)	0.517	0.942 (0.664–1.337)	0.738
Oh et al. [17]	2021	1	Observational study	55.0	II & III	761	634	1.000 (0.776–1.284)	0.989	1.075 (0.869–1.333)	0.500
					II	470	274	0.471 (0.249–0.890)	0.021	0.733 (0.434–1.238)	0.245
					III	291	360	1.045 (0.792–1.379)	0.756	1.186 (0.921–1.527)	0.186

PSM, propensity score matching; S-1,tegafur/gimeracil/oteracil; CAPOX, capecitabine plus oxaliplatin; OS, overall survival; HR, hazard ratio; CI, confidence interval; DFS, disease-free survival.

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
