# Peer review of "Efficacy of S-1 or Capecitabine Plus Oxaliplatin Adjuvant Chemotherapy for Stage II or III Gastric Cancer after Curative Gastrectomy: A Systematic Review and Meta-Analysis"

_cancers, 2022, doi:10.3390/cancers14163940_

Round 1

Reviewer 1 Report

The authors have addressed my comments and I am happy with the resulting manuscript

Reviewer 2 Report

Dear authors:

The authors have revised their manuscipt according to our suggestion. I think this work can be publicated now. 

This manuscript is a resubmission of an earlier submission. The following is a list of the peer review reports and author responses from that submission.

Round 1

Reviewer 1 Report

The authors performed a meta-analysis on the adjuvant use of S1 and CAPOX in gastric cancer

They identified 3 clinical studies 2 propensity score matching and 1 retrospective observational. All studies were performed in Korea.

There is no report on the details of the schedules and compliance. Also, short- and long-term toxicity in relation to the above-described studies are not presented.

A few recent studies have indicated that the response to anti-cancer treatment maybe linked to specific population related genetic variants. There is also recent indication in relation to the role of tumor tissue molecular profiling in the design of personalized treatment not only in the metastatic but also in the adjuvant setting of gastric cancer.

This meta-analysis indicated similar effect of S1 and CAPOX on the survival but several inherent limitations are associated with such type of studies

Based on the above comments, the conclusion that these findings can serve as a basis for the future prospective randomized controlled trial is too generic and can not apply to the entire population requiring adjuvant chemotherapy for gastric cancer world-wide.

Reviewer 2 Report

The author performed a systemic review and meta-analysis regarding S-1 versus Capecitabine Plus Oxaliplatin as adjuvant chemotherapy in gastric cancer patients. Both S-1 and Capecitabine Plus Oxaliplatin are the standard adjuvant chemotherapy regimens. However, the optimal utility of these regimens remains unanswered. This study provided a clinical implication of these two regimens for patients with gastric cancer receiving adjuvant chemotherapy. Nonetheless, there were something need to be improved.

1. This study focused on adjuvant chemotherapy for patients with stage II or III gastric cancer receiving radical surgery with D2 lymph node dissection. The title should be more specific for certain patients. Please revise your title more clearly about what kind of patients investigated.

2. This meta-analysis only enrolled 3 retrospective study. As we know, 2 phase III randomized studies, Classic study and ACTS-GC study, had reported their 5 year survival results. Thus, this meta-analysis should better enrolled these 2 pivotal study for analysis.

3. Besides efficacy, the safety profile is also another important issue. The author should also compare the safety profiles of S-1 with Capecitabine Plus Oxaliplatin. This will be a clinical implication for physicians who treated patients with gastric cancer. 

Reviewer 3 Report

This paper reports about comparison between S-1 and CapeOx as the adjuvant chemotherapy for curatively resected gastric cancer. There are some problems in this paper. It is afraid that more appropriate method can be taken for comparing S-1 with CapeOx.

1. It is very difficult to understand why the results of randomized clinical trial are not included in this meta-analysis. S-1 in the ACTS-GC trial and CapeOx in the CLASSIC trial can be compared via surgery alone. 

2. There are more options of adjuvant chemotherapy for gastric cancer such as S-1 plus docetaxel (JCCRO-GC07 trial) and S-1 plus oxaliplatin (ARTST 2 trial).

3. Three studies is small for meta-analysis. Disease free survival is an important endpoint for evaluating efficacy of adjuvant chemotherapy as well as overall survival. For analysis of disease free survival, 3-year follow up is enough. The subjects should be expanded for the analysis of disease free survival.

4. Although the authors describe, "The staged OS rates were not presented in two previous papers; therefore, OS was analyzed by the authors in previous papers.", it is difficult to understand how it was analyzed.

5. In the results, the authors decribe, "CAPOX regimen tended to be better in stage II patients without statistical significance (HR 0.65, 95% CI 0.38‐1.11; p = 0.14)", small sample size might decrease the statistical power. Considerint the hararad ratios in the ATTRACTION-2 and CLASSIC trials, HR 0.65 is condierably relevant.